# How Poor Is Aphyllophoroid Fungi Diversity in the Boreal Urban Greenhouses of Eastern Europe?

**DOI:** 10.3390/jof9111116

**Published:** 2023-11-17

**Authors:** Anton G. Shiryaev, Ivan V. Zmitrovich, Stepan A. Senator, Elena N. Minogina, Oleg B. Tkachenko

**Affiliations:** 1Institute of Plant and Animal Ecology, Ural Branch of the Russian Academy of Sciences, 620144 Ekaterinburg, Russia; 2V.L. Komarov Botanical Institute, Russian Academy of Sciences, 197376 Saint Petersburg, Russia; iv.zmitrovich@gmail.com; 3N.V. Tsitsin Main Botanical Garden, Russian Academy of Sciences, 127276 Moscow, Russia; stsenator@yandex.ru (S.A.S.); ol-bor-tkach@yandex.ru (O.B.T.); 4Botanical Garden, Ural Branch of the Russian Academy of Sciences, 620144 Ekaterinburg, Russia; minogina71@mail.ru

**Keywords:** botanical garden, non-native fungi, invasion, pathogen, plant disease

## Abstract

It is generally accepted that mycobiota diversity in urban greenhouses is poorer than in natural ecosystems, but our knowledge on this field of research is fragmentary. Here, we present the results of a long-term study of aphyllophoroid macrofungi (Basidiomycota) forming fruitbodies on non-native sub/tropical woody and herbaceous plants in the greenhouses of Saint Petersburg, Moscow, and Ekaterinburg botanical gardens located in the hemiboreal vegetation subzone of Eastern Europe. Over 20 years of research, fruitbodies of 58 species of aphyllophoroid fungi have been identified. Fungal species that developed on the wooden structures of greenhouses and building materials made of local wood are discussed separately. The list of fungi on non-native substrates is dominated by saprobes (93.1% of total list) as well as mycorrhizal with basidiomata on plants (8.6%). Phytopathogens have the lowest number (7.0%), and ¾ of species are widespread locally. Non-native plants are dominated by native fungal species (78.9%), while the percentage of non-native species is low (21.1%). In the three surveyed cities, the area of the studied greenhouses is 2.8 ha, and not a single species of fungi has been found twice on the same substrate. Half of the identified species are characterized by a single specimen (29 species/50.9%). *Hymenochaete rheicolor* was discovered in Russia for the first time and its known distribution is discussed. Only six (*Antrodia gossypium*, *Hyphodontia arguta*, *Lyomyces sambuci*, *Peniophora cinerea*, *Ramariopsis kunzei*, and *Trechispora farinacea*) local species (10.5%) were collected in all the three cities. The α-diversity of mycobiota (mean number of species per site, Shannon Index, and Menhinick Index) in the Ekaterinburg’s greenhouses is 1.2–3.0 times lower compared to suburban forest parks and old-growth natural forests, while β-diversity (Whittaker Index, Jaccard Index, and Morisita–Horn Index), on the contrary, is 2.1–7.7 times higher. With the plants’ age, the probability of detecting fungi on them increases significantly. In greenhouses, phytopathogenic aphyllophoroid macrofungi are collected on woody plants only, but the probability of their development is not related to the plants’ age.

## 1. Introduction

The study of the diversity and dynamics in various components of urban ecosystems is one of the most important areas of modern research in biology and ecology [1]. Forest fragments that have survived the process of urbanization are a characteristic feature of many large cities and are often represented by unique flora complexes and associated mycobiota [2,3]. The vegetation of botanical gardens, which are an integral part of such forests, is a reflection of urban nature—with potentially widespread non-native species, which coexist with stable local relatives and purposefully introduced flora [4].

With non-native plants, many species of non-native fungi are unintentionally spread into urban settings and botanical gardens, which can cause significant environmental and economic damage [5]. In plantings of non-native plants in botanical gardens, a great number of non-native fungal species are concentrated. For some invasive fungal species, plantings of botanical gardens are a refuge for acclimatization to new environments and a springboard for introduction into the local natural forests [3,6]. Although the marked impact of fungal introductions on ecosystems is recognized [4,7,8], general knowledge of the geographic distribution and distribution patterns of non-native fungi is far from complete [2,9,10], and needs to be constantly updated.

The maximum concentration of non-native plant species is characteristic of greenhouses, where the plants are sub/tropical along with their own number of trophically associated exotic fungal species. Under favorable, but limited conditions, such fungi can spread into local nature causing disease or death of local plants [8,11,12,13,14,15]. Nevertheless, there are very little data on the fungal diversity in greenhouses.

Among various groups of macrofungi developing in the greenhouses of Europe, the bulk of the data relate to a targeted study of the species composition of agaricoid fungi. A number of greenhouses located in the nemoral regions of Central Europe (Austria, Germany, Poland, and Czech Republic) were studied, in which the biodiversity of agaricoid fungi was assessed based on the presence of fruitbodies in the soil [6,7,10,14,16,17,18,19]. It has been demonstrated that the number of fungal species in greenhouses is significantly less compared to natural conditions, and mainly local fungal species develop, and the proportion of non-native sub/tropical fungi varies in the range of 10–20% of the total number of species. Agaricoid fungi in greenhouses are humus saprobes that have been sporadically found to be capable of exhibiting pathogenic or mycorrhizal activity [19,20]. The total number of agaricoid fungi in the greenhouses of these countries is 187 species.

Due to the small number of other groups of macrofungi in the greenhouses of Europe, they are rarely the objects of special research. For example, in the greenhouses of the Botanic Garden of the Institute of Botany in Graz (Austria), there are certain isolated data on the findings of aphyllophoroid fungi on wooden structures and building materials, but very few species of fungi have been collected on living or dead parts of non-native plants. In the sub/tropical greenhouses, on the outside of a very moist flower pot with substrate and the seedling of a date palm species, the poroid *Rigidoporus lineatus* (Pers.) Ryvarden was collected [7]. It was also collected in the Budapest greenhouse. This is a common polypore in the tropics, but rare in Europe [12,15]. Basidiomata of the saprobe/pathogen *Schizophyllum commune* Fr. were collected, too [7]. It is a common cosmopolitan decomposer of wood on rather dry, undecayed logs and stumps. This one was also collected in the Czechish greenhouses (Paskow) [11]. In the Warsaw tropical greenhouses, on a dead standing trunk of *Washingtonia filifera* (Linden ex Andre) H.Wendl, the cyphelloid saprobe *Henningsomyces candidus* (Pers.) Kuntze, was found—a native species widely spread in Europe as well as in North Africa, Asia, Australia, North America and South America [10]. Also, the corticoid saprobe *Trechispora farinacea* (Pers.) Liberta, was collected on a dead part of the trunk of *Cyathea australis* Domin [10,19]. This fungal species is native and very common in all the European countries, in Asia, Australia, North and South America [15]. In general, the lists of aphyllophoroid fungi in the greenhouses of individual cities include only 2–6 species, which is 5–14 times less than the lists of agaricoid fungi. Such low results are of interest due to the fact that aphyllophoroid fungi are among the most active wood destroyers in natural ecosystems [13,15].

In Eastern Europe, there have been rare studies of macrofungi in greenhouses. In Saint Petersburg, located in the boreal zone, in the greenhouses of the Peter the Great Botanical Garden on non-native woody plants, on the soil under them, as well as on building materials, structures, fences, shelves made of local woody plants, mainly local species of agaricoid, aphyllophoroid and gasteroid macrofungi were found in 1920–1930 [21,22,23,24,25,26,27]. The list of species is dominated by native fungal species, while non-native taxa are rare. At the beginning of the 21st century, in the city of Ekaterinburg, in the greenhouses of the Botanical Garden of the Ural Branch of the Russian Academy of Sciences, a number of aphyllophoroid fungi were collected on non-native trees [28,29,30,31]. In general, the identified species composition of macrofungi in the greenhouses of these two cities to 2010 was extremely poor. For example, seven species of aphyllophoroid fungi on non-native plants in Saint Petersburg and six species in Ekaterinburg were identified. However, even at the beginning of the 20th century, it was assumed that despite the small lists of fungal species in individual greenhouses, when summing up data from many greenhouses in Saint Petersburg, the total number of species may not be inferior to the species richness in natural conditions, when comparing similar areas of greenhouses and natural conditions [23].

According to the published results for the greenhouses of Central Europe, the identification of fungal fruitbodies is random, many species are represented by single specimen, and fungal diversity in greenhouses is lower compared to natural conditions [7,10,19]. However, the articles do not indicate what diversity level (alpha, beta or gamma) was investigated. In this study, we test the hypothesis that fungal diversity in greenhouses is poorer compared to natural conditions.

In order to find answers to the problems described above, over the last 20 years, our team has been conducting research on the taxonomic diversity of aphyllophoroid fungi in greenhouses in three cities of the boreal part of Eastern Europe: Saint Petersburg (the largest complex of greenhouses is located in the Peter the Great Botanical Garden), Moscow (Tsitsin Main Botanical Garden) and Ekaterinburg (several glasshouses including Botanical Garden of UrB RAS). These greenhouses are of different ages and are also located in different sectors of the continentality of the boreal climate: from the submaritime sector in Saint Petersburg to the subcontinental one in Ekaterinburg.

The purpose of the study is to establish the number of aphyllophoroid fungi species forming fruitbodies on non-native woody and herbaceous plants in greenhouses of Saint Petersburg, Moscow and Ekaterinburg. The answer to this question will enable us to estimate how poor aphyllophoroid fungal diversity in the boreal urban greenhouses compared to natural condition. The following questions will also be explored: Do native or non-native fungal species, saprobes or pathogens predominate? Are non-native or native plant pathogens predominant? What species of fungi develop on structures made of local wood? Are there non-native fungal species on them? Is the number of fungal species related to the size, age of greenhouses, age or size of plants?

## 2. Materials and Methods

### 2.1. Study Area

The studied greenhouses are located in three cities of Eastern Europe: Saint Petersburg, Moscow, and Ekaterinburg (Figure 1). The city of Saint Petersburg (60°00′ N, 30°18′ E; 12 m a.s.l.) has an area of 1439 km^2^ with a population of 5.6 mL. The city is located at the border of south boreal and hemiboreal vegetation subzones [32], with submaritime climate, annual average precipitation 662 mm/year, air humidity 78%, and average annual air temperature 6.6 °C [33]. Over the past 40 years, the average annual temperature in the city has increased by 1.6 °C (in 1960–1990, it was 5.0 °C, and in 2000–2022, it reached 6.6 °C). The city of Moscow (55°45′ N, 37°36′ E; 165 m a.s.l.) has an area of 2561 km^2^ and a population of 13.1 mL. Located in the hemiboreal vegetation subzone, the annual average precipitation is 708 mm/year, air humidity 78%, and the average annual air temperature 6.7 °C. Over 40 years, the average annual temperature has increased by 1.6 °C (from 5.1 °C to 6.7 °C). The area of Ekaterinburg city (56°48′ N, 60°35′ E; 280 m a.s.l.) covers 568 km^2^, whereas the population reaches 1.6 mL. The city is located at the border of south boreal and hemiboreal vegetation subzones, on the border between Europe and Asia. The average amount of precipitation is 562 mm per year, and the average air humidity is 71%. The average annual air temperature is 3.6 °C, and has increased by 1.5 °C over 40 years (from 2.1 °C to 3.6 °C). The climate is continental with a rather characteristic sharp variability of weather conditions and well-defined seasons [33].

Thus, the three studied cities are located in the boreal natural zone, but in different sectors of the climate continentality: from the mild submaritime in Saint Petersburg, through Moscow and to the subcontinental in Ekaterinburg with long and harsh winters.

### 2.2. Description of Greenhouses in the Three Cities

In Saint Petersburg (SPB), the study was carried out in the largest complex of greenhouses located in the territory of the Peter the Great Botanical Garden. The area of these greenhouses is 1 ha (Table 1, Figure 2). In Moscow (MSC), the largest complex of greenhouses located in the territory of the Tsitsin Main Botanical Garden, where the area of greenhouses is 1 ha, was studied. In Ekaterinburg (EKB), a number of greenhouses located in different parts of the city were studied. The largest area of greenhouses is located in the territory of the Botanical Garden of the Ural Branch of the Russian Academy of Sciences (0.3 ha), also in the territory of two city Arboreta (0.14 and 0.12 ha), and also in the Botanical Garden of the Ural Federal University (0.05 ha). In total, the studied area of greenhouses in Ekaterinburg is 0.6 ha.

The average age of greenhouses in the Peter the Great Botanical Garden (SPB) is 110 years, 40 years in the Tsitsin Main Botanical Garden (MSC) and 53 years in Ekaterinburg (EKB) [34,35,36] (Table 1).

The composition of plants in the greenhouses of EKB has been studied in detail. All considered plants are divided into two groups—woody and herbaceous. Woody plants are subdivided into gymnosperms, angiosperms and arborescent. Herbaceous plants are subdivided into evergreen perennials, ferns and annual grasses. The study takes into account plants above 0.5 m in height. The age of each plant is also established. The results of plant accounting were averaged over 10-year age classes: 0–9, 10–19, etc., until 110–119 years (oldest age class). Latin names of plants and names of taxon authors are taken from the Plants of the World Online database [37].

### 2.3. Anthropogenic Gradient

Due to the fact that the study examines the issue of “poorness of the mycobiota in greenhouses”, its testing was carried out in EKB. The anthropogenic gradient was studied starting from the territory with the maximum proportion of non-native plants, i.e., from greenhouses, in which 100% of the plants are non-native to the local flora. Further, the proportion of non-native plants decreases in the Botanical Garden of the Ural Branch of the Russian Academy of Sciences, plantations in the center of EKB, suburban forest parks, and there are no non-native plants in old-growth forests in the vicinity of the city. As regards old-growth forests, the Natural Park “Bazhov’s Tales” located 30 km from EKB was used. The Botanical Garden, the center of Ekaterinburg and its environs are located in forests dominated by Scotch pine (*Pinus sylvestris* L.), the same as within Ekaterinburg. The duration of the study was 20 years (2003–2023). Fruitbodies of aphyllophoroid fungi developing on woody and herbaceous local and non-native plants, saprobes, mycorrhizal and phytopathogens were collected.

Thus, mycobiota of 5 plots were studied along the anthropogenic gradient:Greenhouses: A total of 8 greenhouses in EKB were investigated, the total area was 0.61 ha. The average area of greenhouses was 0.08 ha (from 0.05 to 0.12 ha). Thus, a site of 0.08 ha was used as a model area. A total of 8 sites were studied in greenhouses. Outside the greenhouses, in forests and parks, we also studied sites with an area of 0.08 ha, i.e., radius of 16 m.Botanical Garden of the Ural Branch of the Russian Academy of Sciences: 20 sites, 0.08 ha each.City center of EKB: 20 sites, 0.08 ha each.Suburban forest park: 20 sites, 0.08 ha each.Old-growth forests in the vicinity of EKB: 20 sites, 0.08 ha each.

Therefore, a total of 88 sites were studied, 0.08 ha each.

### 2.4. Mycological Research

The fruitbodies of fungi found only on trees (live and dead, standing and fallen) and roots were taken into account. Fungal species that formed fruitbodies on the soil were excluded from the study. The term “specimen” means the detection of one fungal species on a substrate (log, trunk, or branch) regardless of the number of fruitbodies that make up such a specimen of the species found [38]. The specimen could consist of several fruitbodies. Species richness of mycobiota per area unit was set as the average number of species per hectare (species/ha). All analyzed fungal specimens were collected by the authors of this article in the period of 2003–2023. We have added one species to the list (*Flaviporus brownii* (Humb.) Donk) represented by one specimen which was collected in the greenhouses of the Peter the Great Botanical Garden (SPB) in the 1930s by Prof. A.S. Bondartsev (depositing in LE F), but we did not find this species in the SPB greenhouses.

In all greenhouses, fungi were collected using the route method [38] with the study of the maximum possible number of plants. In order to study the hypothesis of how poor the mycobiota diversity is in greenhouses, fungal specimens were collected within 88 sites located in EKB greenhouses and model sites in Ekaterinburg city center and forested areas in the suburbs (see Section 2.3). For specimen collection, the route method was used on average 4 times during year over the past 20 years. The maximum range of tree and shrub substrates located inside the sites was studied. The fruitbodies of fungi developing on living trees were classified as pathogens, but if fruitbodies were collected on dead branches and trunks, then they were classified as saprobes. Mycorrhiza-formers included species classified according to UNITE (https://unite.ut.ee/ (accessed on 23 June 2023)).

All fungal specimens collected by the authors were identified based on the morphological features of fruitbodies and according to the following key books [13,15,27,39], using light microscopes LEICA 2000 (Wetzlar, Germany) and ZEISS Axio Imager A1 (Jena, Germany). In our studies, we adhered to a classical morphological approach, based on the idea that the development of fruitbodies marks the accumulation of mycelium biomass within the substrate, i.e., predominance of this species in the corresponding substrate entity. We agree that molecular screening can reveal greater species diversity per substrate entity, but it often also reveals inactive propagules and ephemeral mycelia of low competitive potential, i.e., incapable of its complete colonization [40]. The collected specimens are deposited in the Institute of Plant and Animal Ecology Herbarium (SVER) and Komarov Botanical Institute Herbarium (LE F). The fungal species nomenclature is given according to Index Fungorum [41].

All fungal species were subsumed under one of the two categories: native or non-native. Non-native species were defined as species untypical of the local mycobiota, whose introduction into a given area was not associated with the natural course of mycogenesis, but resulted from direct or indirect human activity [2]. The main source used to distinguish between native and non-native species was *Aphyllophoroid fungi of Sverdlovsk region* [31]. The GBIF data on the worldwide distribution of fungal species (www.gbif.org; accessed on 19 April 2023) were used.

### 2.5. Statistical Analysis

In order to draw a dendrogram reflecting the similarity of the species composition of fungi along the anthropogenic gradient, the Ward method and the Pearson correlation coefficient were used. The Spearman’s rank correlation coefficient (r) was used to establish a linear correlation between various parameters of mycobiota, climate, and plants.

The α-diversity of mycobiota was estimated in three ways: as the average number of fungal species in five plots along anthropogenic gradient as well as by the value of the Shannon index, and the Menhinick index. The β-diversity was also assessed in three ways: the average Jaccard index as well as the Morisita–Horn Index and the Whittaker Index [38]. Statistical differences between the sites along the anthropogenic gradient were established using the Mann–Whitney (U-test) method.

An approach based on the sampling algorithm was used to estimate the expected number of species per transect [42]. This approach is based on a rarefaction curve drawn using a special algorithm for random multiple permutation of data within samples from among the detected samples. An indirect method for assessing the completeness of the identification of species richness was also used—the Turing coefficient (C), which was based on the ratio between the number of singleton species (represented by a single find) and the total number of identified species [43].
C = (1 − f_1_/S) × 100%,(1)
where f_1_ is the number of singleton species, and S is the total number of identified species. The potential number of species (T) can be calculated as
T = S/C(2)

Statistical data processing was carried out using statistical software packages Statistica 10.0 and MS Excel 2007.

## 3. Results

### 3.1. List of Fungal Species in the Greenhouses of the Three Cities

During this study, 140 samples representing 58 species of aphyllophoroid macrofungi (Appendix A) were collected in the greenhouses of the three cities (Table 2). Most specimens and species were collected in the greenhouses of EKB (84 specimens/39 species), then in the Main Botanical Garden of MSC (39/28), and to a lesser extent in the Peter the Great Botanical Garden of SPB (17/17).

An amount of 46 species (79.3% of the total) are local, widespread in the boreal part of Eastern Europe, while 12 species (20.7%) have range centers in other biogeographic regions of the planet. A number of non-native species at the time of their collection were new to Russia. For example, in the 1920s, it was *Flaviporus browni*, and, in the 2000, *Physalacria cryptomeriae*. In this paper, *Hymenochaete rheicolor* is reported in Russia for the first time. In the greenhouses of SPB, one species can be classified as non-native (5.9% of the number of species in SPB), in MSC, three species (10.7%), and, in EKB, eight (19.5%). Thus, as the number of identified fungal species grows, the proportion of non-native species increases linearly.

Poroid fungus *Flaviporus brownii* is non-native to Eastern Europe, collected in the palm greenhouse of SPB [27] and widely spread in nemoral and subtropical Europe, sub/tropics of America, Asia, Africa, Australia and New Zealand (https://www.gbif.org/ru/species/5954717 (accessed on 19 June 2023)).

In MSC, a non-native poroid *Lindtneria trachyspora* was collected having a mode of finds in nemoral and subtropical Europe and eastern North America; there were finds in the tropics of Central and South America (Brazil, Costa Rica, El Salvador, Mexico), Africa (Seychelles, South Africa), in Australia and East Asia (Japan) (https://www.gbif.org/ru/species/2552491 (accessed on 19 June 2023). Stereoid fungus *Hymenochaete rheicolor* has a mode of findings in the sub/tropical regions of America (Argentina, Brazil, Chile, Costa Rica, Cuba, Dominican Republic, Ecuador, French Guiana, Mexico, Panama, Puerto Rico, Jamaica, and the USA), Asia (China, India, Nepal, Philippines, Japan, and Thailand) and Australia (https://www.gbif.org/en/species/2519646 (accessed on 19 June 2023). *Thanatephorus terrigenus* is a European species known in Germany, Great Britain, Norway, Sweden (https://www.gbif.org/ru/species/2555158 (accessed on 19 June 2023) and the Urals [31].

In the EKB greenhouses, the poroid *Microporus xanthopus* was collected on the wall of a box made of the wood of the East Asian *Carpinus cordata* Blume, in which the *Magnolia obovata* Thunb. seedling had been brought from the Russian Far East [44]. Under the natural conditions of Russia, this fungus was found only in the south of the Far East in Primorsk Territory [45]. East Asian clavarioid fungi *Physalacria cryptomeriae* and tropical *Ph. orientalis* were collected on dead branches of *Cryptomeria japonica* (Thunb. ex L.f.) D.Don. and *Magnolia grandiflora* L., respectively. At the base of *Eucalyptus* sp. and on the wall of a clay tub, a poroid *Rigidoporus lineatus* was collected—widespread in tropical regions, but also often found in temperate greenhouses. An interesting finding was the corticoid *Leptosporomyces raunkiaeri* known in Leningrad [46] and Nizhny Novgorod provinces of Russia (https://www.gbif.org/ru/species/2554208 (accessed on 19 June 2023). This fungus was recently found in Sverdlovsk province on a dead tree of *Alnus glutinosa* (L.) Gaertn. in a suburban forest of EKB [47]. The question arises: did this fungus inhabit the natural biocenoses of the region and from there “come“ to the greenhouses, or vice versa—was it brought into the greenhouses, and is it now settling into nature?

In the three cities studied, there is no correlation between plant height and the number of identified fungal species (r = 0.44, *p* = 0.37). It was also found that the number of fungal species did not depend on the area of greenhouses (r = –0.85, *p* = 0.31) nor on the average age of greenhouses (r = –0.86, *p* = 0.09). At the same time, in the EKB greenhouses, a strong positive linear regression (Figure 3) was found between the level of sampling effort (as the distance traveled when detecting fungi in greenhouses) with the number of detected fungal specimens (r = 0.99, *p* = 0.0001) as well as polynomial fit (3 order) between the sampling effort with the number of fungal species (R^2^ = 0.96, *p* = 0.0002).

With the current number, age and condition of plants in EKB greenhouses, 52.1 ± 4.8 species of aphyllophoroid fungi can be identified (Figure 4) with 39 species collected by us (75% of the potential species composition). Consequently, the revealed diversity is currently far from the potentially possible one. At the same time, 12 out of 39 species of aphyllophoroid fungi in the EKB greenhouses are characterized by a single find (singletons); therefore, the Turing coefficient is 69%, which indirectly confirms the estimated species richness of 56 species.

Six species of aphyllophoroid fungi (10.5% of the total number) were collected in the greenhouses of all the three cities: *Antrodia gossypium*, *Hyphodontia arguta*, *Lyomyces sambuci*, *Peniophora cinerea*, *Ramariopsis kunzei*, and *Trechispora farinacea*. All these species were saprobes. Between the SPB and MSK greenhouses, the Jaccard similarity index is 0.27, between SPB and EKB, it is 0.11, and, between MSK and EKB, it is 0.52.

Four species (6.9%) are phytopathogens (facultative). Three of them (*Helicobasidium purpureum*, *Thelephora palmata*, and *Th. terrestris*) are local species widespread in nature and greenhouses. One species, *Athelia rolfsii*, is non-native, with a mode of finds in the sub/tropical regions of the planet, but has long been widely distributed in greenhouses of the temperate zone (https://www.gbif.org/ru/species/2554082 (accessed on 23 June 2023)). Each of the four species of phytopathogens was collected in the greenhouses of only one city.

There were 115 specimens collected on woody plants (82.1% of total), which comprised 50 fungal species (Table 3). These were mainly saprobes (42 species/102 specimens) and mycorrhizal (4/9). All phytopathogenic fungi were collected on woody plants only. An amount of 25 specimens of 10 species were collected on herbaceous plants. All species on herbaceous plants were saprobes. On all plants, the largest number of specimens (90.7%) belonged to saprobes, 6.4% to mycorrhizal, pathogens were represented by 2.8% of the specimens. Fungi were not detected on herbaceous annual plants. Consequently, the number of fungal specimens collected on woody plants exceeded the number on herbaceous ones in SPB by 16 times, in MSC by 5.3 times, in EKB by 3.4 times, and the average for the three cities was 4.3 (Table 2). An amount of 88.9% of mycorrhizal fungi were collected on angiosperms, while 52% of saprobes were collected on angiosperms, and 21.3% on arborescent.

In the three surveyed cities, during 20 years, no fungal species was found twice or more times on the same plant species. Moreover, not a single fungal species has been found twice or more times on the same genus or even family of plants.

### 3.2. Fungal Diversity on Model Plots in Ekaterinburg along the Anthropogenic Gradient

Taking the example of the EKB greenhouses as the best studied ones of the three analyzed cities (the maximum sampling effort was applied, the most specimens and species of fungi were collected), it is possible to study fungal diversity along the anthropogenic gradient: from greenhouses, with a 100% set of non-native plants, through the Botanical Garden UrB RAS, the center of EKB, suburban forest parks and old-growth forests located 40 km from Ekaterinburg, where only native plants grow (Table 4).

Three studied parameters of mycobiota α-diversity are lower in greenhouses compared to natural conditions. For example, the average number of fungal species within a site is 3.0 times lower compared to natural conditions, the Menhinick index is 1.3 times lower, and the Shannon index is 1.2 times lower. On the other hand, all three parameters reflecting the β-diversity of mycobiota are higher in greenhouses: the Whittaker index is 2.1 times higher, the Morisita–Horn index is 7.5 times higher, and the Jaccard similarity index is 7.7 times higher. Significant statistical differences (Mann–Whitney U–test) were detected for average number of fungal species per plot (U = 0, Z = –3.36, *p* = 0.0008) between the greenhouses and old-growth forest, as well as for Jaccard Index (U = 0, Z = –3.48, *p* = 0.00007). The Shannon Index differed significantly between greenhouses and old-growth forests (*p* < 0.01), with higher diversity in old-growth forests compared to the greenhouses.

Along the anthropogenic gradient from natural forests to the greenhouses, where the proportion of non-native species of flora increased from 0% to 100%, and the average height of stands decreased from 13.1 m to 2.6 m, the number of non-native fungal species grew from 0 to 8, while the proportion of non-native fungal species increased from 0% to 20.5%, the density of non-native species grew from 0 species to 13.1 species/ha.

A comparison of the lists of fungal species in the five studied plots indicates that suburban and old-growth forests are clustered together, forest plantations in the city center are combined with the Botanical Garden, and the mycobiota in greenhouses is significantly different (Figure 5).

In 2022, 5748 plant specimens (above 0.5 m) were counted in the EKB greenhouse complex, of which 2167 were woody plants and 3581 were herbaceous. The oldest plants are over 110 years old, but young individuals predominate—a third of the total numbers of plants are aged from 0 to 20 years (Figure 6A). For all plants, as well as separately for woody and herbaceous plants, a strong positive relationship was found between the age and the number of fungal specimens developing on them (R^2^ = 0.99). In all cases, the relationship was S-shaped (Figure 6B): no fungi were found on young plants (0–20 years old), and some fungi were found on 1–3% of plants aged 30–50 years only. Further, on plants aged 60–90 years, the proportion of plants increased exponentially from 4–11% to 74–86%. As a result, the dependence reached a plateau—the fruitbodies of aphyllophoroid fungi were collected on all plants (100%) over 100 years old.

## 4. Discussion

### 4.1. Aphyllophoroid Fungi in Greenhouses of the Three Boreal Cities

In the greenhouses of three boreal cities, on living and dead parts of non-native plants, local species of fungi predominate, including 46 species, which is almost 80% of the total list. They also predominate in terms of the number of samples (127) accounting for 90.7% of the total number of samples. There are significantly fewer non-native species—12, the findings of which represent single specimens.

More specimens of fungi have been collected on woody plants. In SPB, out of 17 samples, 16 were collected on woody plants, in MSC, 32 out of 38 were collected on woody plants, and in EKB, 65 out of 85 were collected on woody plants. As the sampling effort increases, the proportion of species on woody versus herbaceous plants decreases from 16.0 in SPB to 5.3 in MSC and to 3.4 in EKB.

It is worth noting that in the greenhouses of Central Europe (Austria, Germany, Poland, and Czech Republic), the number of aphyllophoroid fungi species varies from three to eight species only [6,7,10,11,14,16,17,18,19,20,39]. A number of species identified in these studies were also collected on non-native plants and in greenhouses of boreal Europe (*Bjerkandera adusta*, *Rigidoporus lineatus*, and *Trechispora farinacea*), while others were absent (*Asterostroma cervicolor*, *Henningsomyces candidus*, etc.). On building structures, boxes and tubs, species were also collected indicated in greenhouses in Eastern Europe: *Antrodia vaillantii*, *Coniophora puteana*, *Hyphodontia quercina*, *Laetiporus sulphureus*, *Schizophyllum commune*, but there are also a number of species not collected by us, for example, *Dentipellis fragilis* (Pers.) Donk, *Hericium coralloides* (Scop.) Pers., *Postia floriformis* (Quél.) Jülich, *Steccherinum ochraceum* (Pers.) S.F. Gray. In Czech greenhouses, the number of fungal species depends on the greenhouse area and the study duration. On the other hand, in the Austrian and Polish greenhouses, it was found that the identified number of fungal species over a period of 3–4 years no longer increased with repeated studies, which could indicate the identification of a complete list [10,11,19].

In contrast to the Central European greenhouses, our study did not reveal any correlation between the number of fungal species in greenhouses and their area, which was due to the fact that the greenhouses in SPB and MSC had the largest areas. However, the greatest number of species was collected in EKB, because purposeful studies were carried out here for several years. In our opinion, for the same reason, there is no correlation between the average age of the greenhouses. We also found a strong positive relationship between fungal species richness and plant age, where the strength increased when studying plants aged 70 years and older (Figure 6).

Widespread local species of aphyllophoroid fungi predominate in greenhouses, but Red Data Book and rare species do not develop. We did not find in the studied greenhouses any fungi included in the Federal Red Book of Russia [48] or in the Red Lists of the three studied regions [49,50,51]. Only one protected species was collected in the EKB greenhouses—the poroid *Abortiporus biennis*—which had an IUCN threat category (CR B1; D1) in Sverdlovsk province [31]. This fungus grew in a woody tub at the base of the Madagascar palm *Dypsis lutescens* (H. Wendl.) Beentje and J. Dransf.

Some non-native species of fungi in greenhouses are predominantly distributed in the sub/tropical zones of the planet: *Flaviporus brownii*, *Hymenochaete rheicolor*, *Lindtneria trachyspora*, *Physalacria orientalis*, and *Rigidoporus lineatus*; others are common in European nemoral forests: *Hypochniciellum ovoideum*, and *Thanatephorus terrigenus*; but four are common in East Asia: *Clavulina ornatipes*, *Clavulinopsis aurantiocinnabarina*, *Microporus xanthopus*, and *Physalacria cryptomeriae*. It is possible that this distribution of new species reflects the centers, from which the plants were brought to the greenhouses. We know that in the early 1950s, some plant seedlings were brought to the Botanical Garden UrB RAS (EKB) along with soil from the Russian Far East, which can explain the findings of East Asian fungal species here. Each of the non-native species is collected in one city only. The proportion of non-native species in greenhouses has a strongly positive relation to the number of fungal species identified and selective effort. Undoubtedly, there are still few studied points to prove this result, but for the future, it will be interesting to test on more material. Is this dependence characteristic of aphyllophoroid fungi in the boreal zone only, or does it also work in the nemoral zone?

In the three cities, only four species of pathogenic fungi were collected, which was unexpected considering the old age of many plants as well as the studied group of macrofungi—aphyllophoroids, which was the most common and numerous of phytopathogens in boreal forests [8,27]. It should be noted that pathogenic species of agaricoid fungi are absent in the greenhouses of Central Europe [7,10,11,19]. This is due to the regular fungicidal treatment of plants and the removal of diseased plants or dead plant parts. All pathogens (including facultative) collected in greenhouses in Eastern Europe (*Athelia rolfsii*, *Helicobasidium purpureum*, *Thelephora palmata*, and *Th. terrestris*) do not cause dangerous diseases in adult plants, although some species from the genus *Thelephora* can cause mass death of young seedlings of coniferous plants [25]. Three of the four phytopathogenic macromycetes are native species, and only one (*Athelia rolfsii*) has a center of finds in the sub/tropics of the planet, but is widely spread in the greenhouses of cities in the temperate zone.

In the Central European greenhouse complexes of Bayreuth, Graz, Jena, Paskov and Warsaw, the smallest number of agaricoid species was collected in Warsaw (16), and the maximum in Bayreuth (79) and Paskov (88). For example, out of 16 species found in Warsaw, 50% were found in other greenhouses. Overall, the average Jaccard similarity coefficient of the agaricoid fungi species composition between these five greenhouses is 0.41. In Central Europe, some tropical species have been first introduced into greenhouses, some of them later spreading in the wild, e.g., *Leucocoprinus birnbaumii* (Corda) Singer, *Gymnopus luxurians* (Peck) Murrill as well as species new to the Old World, e.g., *Melanotus flavolivens* (Sacc.) Singer, *Hydropus fluvialis* Singer, and *Phloeomana neospeirea* (Singer) Gminder [7,16]. The large number of species found in greenhouses of Bayreuth Botanical Garden may result from such factors as the stability of climatic conditions throughout the year and the relatively high diversity of the habitats at this location [6]. In the greenhouses of the Botanical Garden in Warsaw, a small number of species can also be associated with unstable climatic conditions (fogging occurs only in summer in tropical and succulent greenhouses, and the buildings are not heated during the summer months). The authors also note the relatively young age of trees and shrubs, a small amount of dead wood and thorough care of plants can limit fungal species diversity. It turns out that in humus saprobes, which form the basis of the list of agaricoid fungi, the similarity between greenhouses is below 50%, and about 80% are local species, but there are also non-native ones. These parameters are similar to those found by us for saprobic (wood-destroying and litter) aphyllophoroid fungi.

Inside boreal greenhouses on wooden foundations, shelves, tubs, fences, stumps, etc., only local fungal species were collected, because the listed substrates were made of local trees. In all the greenhouses, it is mainly pine, spruce and willow wood. Fungi developing on these substrates are similar for the three cities. The difference in the list of fungi lies only in the substrates available in SPB and MSC—in these cities, many structures are made of oak, while in EKB, there are no oak structures, tubs, shelves, foundations, or beams due to the fact that oak is absent in the natural conditions of the Middle Ural.

The most common fungal species in all the three cities are: *Fibroporia vaillantii* (DC.) Parmasto, *Hydnomerulius pinastri* (Fr.) Jaroch et Besl, *Laetiporus sulphureus* (Bull.) Murrill, *Neoantrodia serialis* (Fr.) Audet, *Serpula lacrymans* (Wulfen) J. Schröt., and *Schizophyllum commune*. Common for SPB and MSC, but absent in EKB: *Gloeophyllum trabeum* (Pers.) Murrill, *Heterobasidion annosum* (Fr.) Bref., *Hymenochaete rubiginosa* (Dicks.) Lév., *Rigidoporus sanguinolentus* (Alb. et Schwein.) Donk. Common between SPB and EKB: *Gloeophyllum odoratum* (Wulfen) Imazeki, *Hyphodontia spathulata* (Schrad.) Parmasto, *Peniophora pini* (Schleich. ex DC.) Boidin, *Rhodofomes roseus* (Alb. et Schwein.) Kotl. et Pouzar.

The similarity between mycobiota on non-native plants in the greenhouses of the three surveyed cities is low compared to the natural mycobiota of Leningrad, Moscow, and Sverdlovsk provinces (0.30 vs. 0.68) [31]. In the three surveyed cities, no fungal species was found twice on the same substrate, which indicates the dynamic nature of the urban mycobiota compared to natural environments. Only six species were collected in the greenhouses of the three cities, which is 4.7 times lower than the natural mycobiota of three regions. Such fragmentation of the species composition of greenhouse mycobiota is also typical on the local level. The average similarity of the species composition between the mycobiota of the greenhouses in EKB is only 0.08, although the distance between the compared areas can be only a few meters. This is explained, among other things, by the fact that half of all fungal species in the three cities are characterized by a single specimen, and others by 2–3 specimens. For 20 years of fungi collecting, we have established places in greenhouses where fungi develop, but about 90% of the species were no longer found on the same substrate. Similar features are characteristic of mycobiota developing in extremely heterogeneous natural and climatic conditions, for example, in ecotone zones like forest-tundra, forest-steppe, in ultracontinental regions and anthropogenic deserts [52].

### 4.2. Fungal Diversity along Anthropogenic Gradient in Ekaterinburg

The average number of species per site in the greenhouses is 3 times lower than in protected old-growth forests. Two other parameters of α-diversity, such as the Menhinick Index and the Shannon Index, which take into account the ratio of the number of specimens and species, turned out to be lower in the greenhouse only by 20–25%, i.e., on the verge of significance (*p* ≤ 0.05). The spatial changes of the β-diversity are opposite compared to the α-diversity. For example, the Whittaker, Morisita–Horn and Jaccard Indexes demonstrate a 2 to 7 times higher β-diversity in the greenhouses in contrast to old-growth forests.

Thus, the 20-year study of α-diversity assessment has two results: on the one hand, the average number of species in the sites is clearly lower compared to old-growth native natural forests. However, when diversity indices are included in the analysis, which takes into account the qualitative and quantitative composition of microbiota, these differences are practically reduced to a statistical error. Regardless of the method of assessment, β-diversity in greenhouses is 2.1–7.7 times higher than in protected natural old-growth forests. Some areas in the greenhouses of EKB do not have a single common view, which confirms the mosaic nature of the transformation of fungal communities and indicates an increase in the spatial isolation of local populations. It is possible that the increase in β-diversity in the gradient “local old-growth forests—greenhouses with 100% composition of non-native plants” was due to the uneven elimination of species in space. This is due to the fact that with an increase in the pessimism of conditions for the local mycobiota in greenhouses, not the actual elimination of species is observed, but a decrease in the number of local populations. Even at the lowest levels of non-native plant phytomass, the most common and abundant native fungal species are able to persist and survive in “habitat fragments” where, for various reasons, conditions can remain relatively favorable. On non-native plants in the greenhouses, local ubiquitous fungi (80% of the total number of species) are collected, the trophic spectrum of which includes numerous local deciduous woody and herbaceous plants. Over many years of development of non-native plants in greenhouses, such fungi have adapted to a new substrate and found a rare opportunity to form basidiomata on dead parts (branches, dead roots, etc.), despite the regular treatment of plants with fungicides, cleaning of dead parts of plants and diseased individuals. Perhaps this can explain why half of the fungal species identified in the greenhouses of the three cities are represented by one sample. Therefore, the hypothesis of low fungal diversity in greenhouses is partially supported by this study: α-diversity is lower or close compared to natural forests, while β-diversity is significantly higher.

In the direction of greenhouses, a sharp increase in the number of non-native fungi, their share in the total species composition, and species density was established. In general, the natural conditions of the Middle Urals do not contribute to the spread of non-native species of macrofungi into nature, although over the past 20 years in EKB, the number of identified non-native fungal species has increased exponentially [53]. Severe cold winters limit the development of non-native thermophilic species of flora and mycobiota. Undoubtedly, some species are capable of spreading through non-native woody plants planted in gardens and forest belts along roads and railways, but the penetration of non-native species of aphyllophoroid fungi into stable old-growth boreal forests has so far been detected only in a few anthropogenic territories [31,47,54,55,56].

The oldest plants in the EKB greenhouses, which are over 110 years old, are *Ficus macrophylla* Pers. (115 years), *Casuarina equisetifolia* L. (110 years), *Strelitzia nicolai* Regel. et K.Koch (111 and 102 years), *Cycas revoluta* Thunb. (101 years), and *Cupressus sempervirens* L. (100 years). The proportion of plants, on which fungal fruitbodies were found was significantly associated with age; however, the relationship was not linear, but S-shaped (Figure 6b). On the most numerous group of plants, which were less than 30 years old, only single fruitbodies of fungi were found, which does not exceed 0.1% of the number of plant individuals. The number of plants aged 30–50 years is 10 times less than the youngest ones, while the number of fungal finds is growing, which makes it possible to increase the proportion of plants with fungi up to 3–5%. The smallest number of plant specimens corresponds to ages over 100 years. At the same time, fungal fruitbodies were collected on all the oldest plants over 100 years old.

On the other hand, the height of non-native plants does not explain the revealed fungal species richness. We explain this by the fact that many woody plants in greenhouses are often heavily pruned as they reach the ceiling. Others in tubs grow very slowly despite their considerable age, although those planted in the ground of the same age stretch very quickly. It should not be forgotten that many herbaceous plants die off to the level of the soil, but their bases are constantly alive and mort-mass is formed around them, on which fruitbodies of litter saprobes develop.

However, despite the fact that, in the nemoral regions of Europe, the results were obtained for agaricoid macrofungi (humus saprobes), while our study was carried out in boreal Europe, mainly on wood-inhabiting and partially pathogenic and mycorrhizal aphyllophoroid macrofungi, some results were similar.In greenhouses, the list of species is poor compared to natural conditions, regardless of the time of study.Low similarity between greenhouses located in the boreal and nemoral parts of Europe. About half of all species are characterized by single finds.Despite the fact that 100% of plants in greenhouses are non-native, no more than 20% of the total lists of non-native saprobes, mycorrhizal and pathogenic macrofungi were collected.The number of fungal species is related to the plants age and selective effort.

## 5. Conclusions

Our results partially confirm the well-known hypothesis that fewer fungal species are known in greenhouses compared to natural conditions [7,21]. It is worth noting that previous studies have not compared the diversity of fungi in greenhouses and nature, given the similar level of sampling effort and the same study area. Our results show that, if we analyze the material obtained from the same study area within a short period of time, for example, 1 day of study or 1 km traveled, then the number of species collected in greenhouses will be lower compared to natural conditions. However, over a longer period of time, 5–20 years, the number of collected fungal species in greenhouses increases, but in natural conditions, changes is often minimal.

In general, during long-term studies of similar area and with the same level of sampling effort applied, the α-diversity of aphyllophoroids in the greenhouse turns out to be lower than in old-growth forests, while β-diversity is 2–7 times higher. Therefore, the hypothesis put to the test is valid for α-diversity only. However, the identified trends are typical of the group under study, or of the boreal nature zone only. Further studies will refine the results obtained.

## Figures and Tables

**Figure 1 jof-09-01116-f001:**
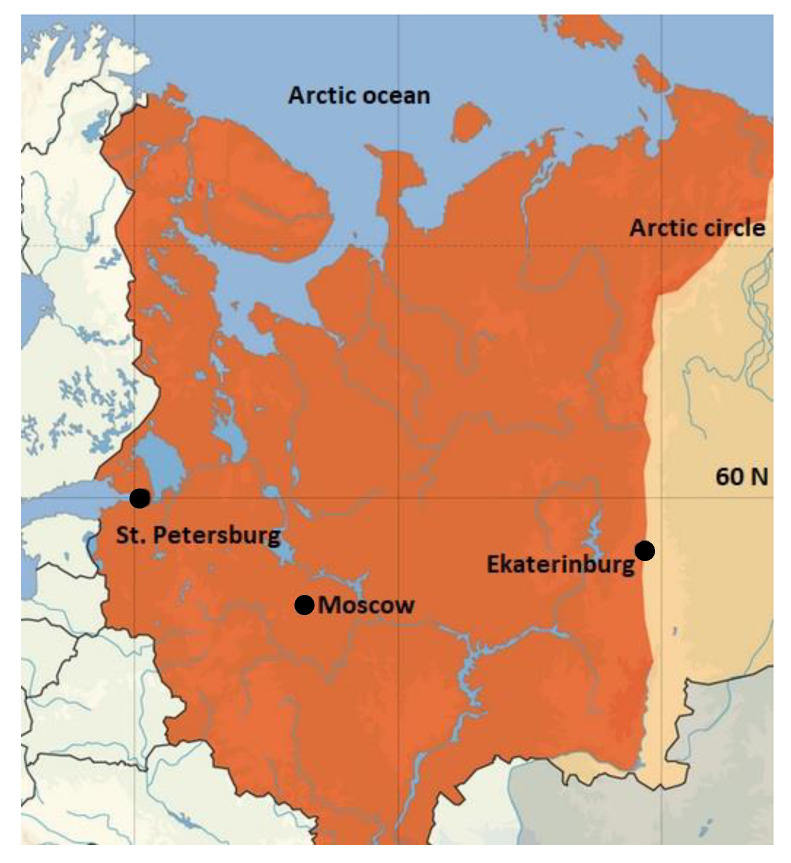
Locations of the three studied cities in Eastern Europe: Saint Petersburg, Moscow, Ekaterinburg.

**Figure 2 jof-09-01116-f002:**
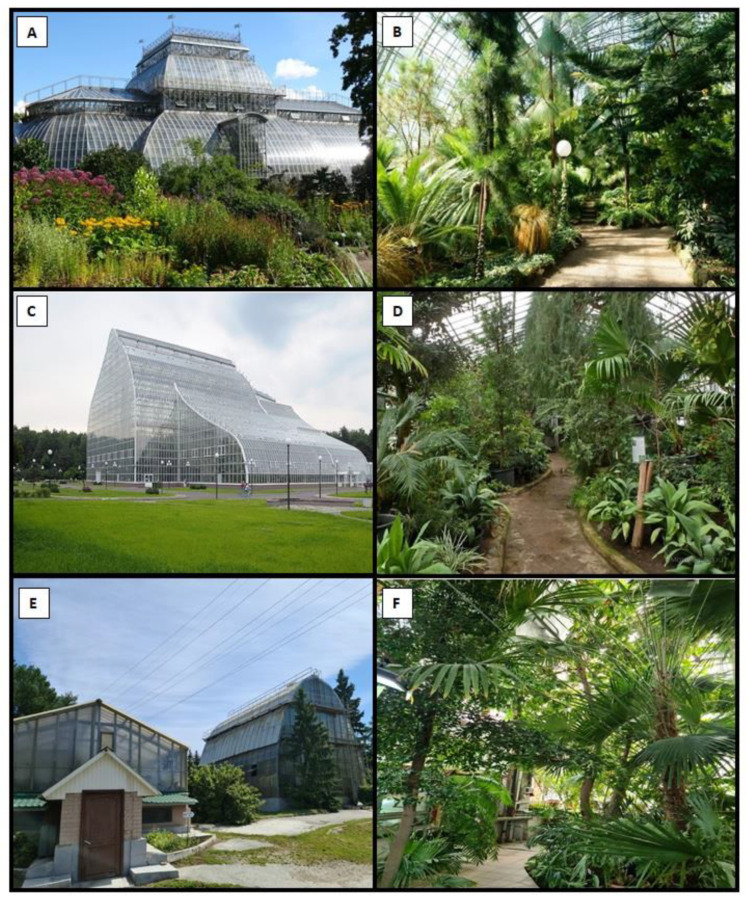
Greenhouses in the studied cities of Eastern Europe. (**A**,**B**)—Peter the Great Botanical Garden (Saint-Petersburg); (**C**,**D**)—Tsitsin Main Botanical Garden (Moscow); (**E**,**F**)—Botanical Garden of Ural Branch of the Russian Academy Sciences (Ekaterinburg).

**Figure 3 jof-09-01116-f003:**
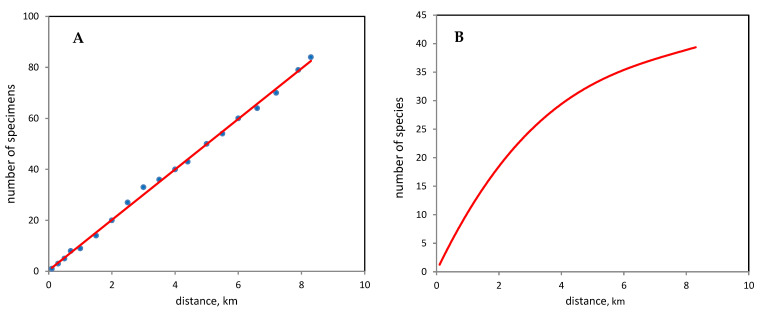
Relationship between sampling effort (distance traveled within the greenhouses) and (**A**) number of fungal specimens (y = 9.89x + 0.38, r = 0.99, *p* = 0.0001) and with (**B**) number of fungal species (y = 0.05x^3^ − 1.24x^2^ +11.54x + 0.11, R^2^ = 0.99, *p* = 0.0002).

**Figure 4 jof-09-01116-f004:**
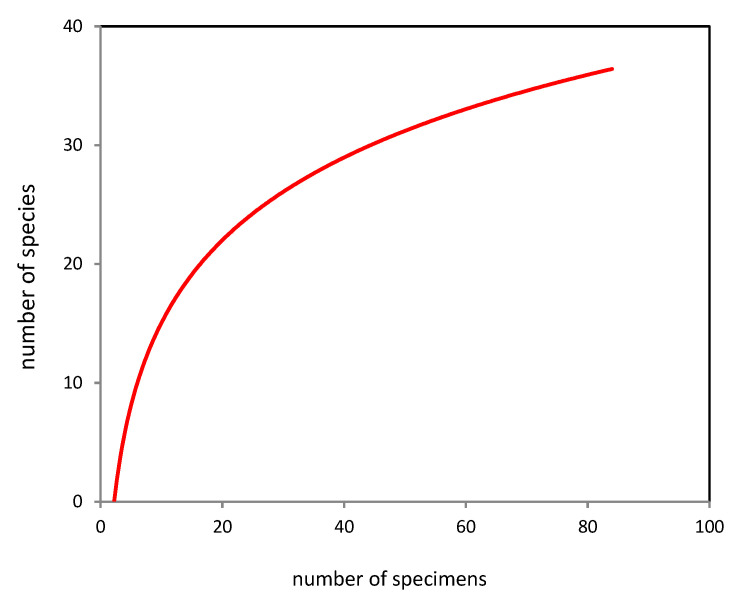
Completeness of identification of aphyllophoroid fungi species in the EKB greenhouses for 20 years, depending on the number of specimens. A smoothed rarefaction curve (individual-based rarefaction curve) is presented, depending on the number of specimens.

**Figure 5 jof-09-01116-f005:**
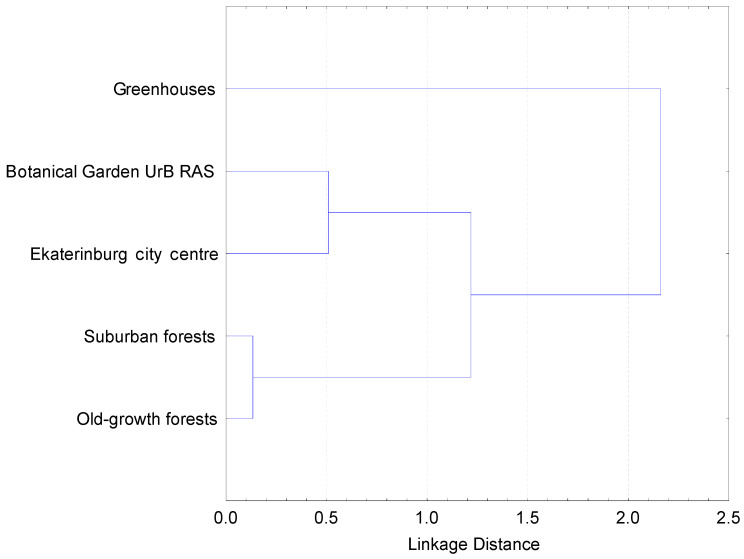
Similarity of the species composition of aphyllophoroid macrofungi in 5 plots located along anthropogenic gradient in Ekaterinburg city, suburban forests and old-growth forests.

**Figure 6 jof-09-01116-f006:**
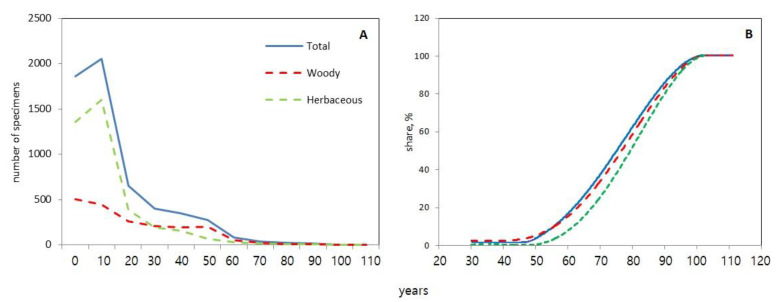
Number of non-native plant specimens (woody, herbaceous and total) in the Ekaterinburg city greenhouses divided into decade groups by decades from the youngest plants of 0–20 years old to the oldest ones of 100–110 years old (**A**); and the percentage of plant specimens, on which aphyllophoroid fungi fruitbodies were found (**B**).

**Table 1 jof-09-01116-t001:** Average parameters for greenhouses in Saint-Petersburg, Moscow and Ekaterinburg.

Greenhouses	Year of Erection	Area, ha	Temperature in Winter/Summer, °C	Humidity in Winter/Summer, %	Major Collections
Peter the Great Botanical Garden (SPB)	110	1.0	12–21/*	**/40–70	*Musa*, *Punica*, *Coffea*, *Camellia*, *Fuchsia*, *Bougainvillea*, *Dicksonia*, *Cyathea*, *Cibotium*, *Fuchsia*, *Plumeria*, *Begonia*, *Agathis*
Tsitsin Main Botanical Garden (MSC)	40	1.0	12–20/*	**/40–80	*Nerium*, *Dicksonia*, *Cyathea*, *Cibotium*, *Cycadopsida*, *Ficus*, *Crinum*, *Hibiscus*, *Begonia*, *Cycas*, *Passiflora*, *Anthurium*, *Pellea*
Botanical Garden of the UrB RAS (EKB)	53	0.61	13–16/*	**/35–70	*Citrus*, *Dicksonia*, *Cyathea*, *Musa*, *Punica*, *Coffea*, *Camellia*, *Fuchsia*, *Bougainvillea*, *Brunfelsia*, *Olea*, *Platycerium*, *Camellia*, *Agave*

Note: *—no heating, indoor temperature depends on external conditions, a system of blackout blinds is used in hot weather; **—no fogging.

**Table 2 jof-09-01116-t002:** Species list of aphyllophoroid fungi in the greenhouses of the three boreal cities (number of specimens).

Species	Trophic Group	City			Information
		SPB	MSC	EKB	
*Abortiporus biennis* (Bull.) Singer	Sap			1	EKB: at the base of *Dypsis lutescens* (H. Wendl.) Beentje et J.Dransf. (Arecaceae) and on woody box.
*Antrodia gossypium* (Speg.) Ryvarden	Sap	1	1	3	SPB: on dead part of deciduous tree/MSC: at the base of *Ginkgo biloba* L. (Ginkgoaceae)/EKB: at the base of *Cryptomeria japonica* (Thunb. ex L.f.) D.Don (Cupressaceae); on *Ficus carica* L. (Moraceae); on dead part of *Eriobotrya japonica* (Thunb.) Lindl. (Rosaceae)
*Athelia bombacina* (Link) Pers.	Sap		1		MSC: on dead part of *Cycas revoluta* Thunb. (Cycadaceae)
*Athelia epiphylla* Pers.	Sap			2	EKB: on dead petioles of *Pteris* sp. (Pteridiaceae); on root neck of *Strelitzia reginae* Banks ex Aiton (Strelitziaceae).
*Athelia rolfsii* (Curzi) C.C. Tu et Kimbr.	Pat			1	EKB: at the base of *Hedera helix* L. (Araliaceae).
*Bjerkandera adusta* (Willd.) P. Karst.	Sap		2	2	MSC: at the base of *Berberis rotundifolia* Poepp. et Endl. (Berberiaceae); on dead stem of *Eleodendron croceum* (Thunb.) DC. (Celastraceae)/EKB: dead branch of *Laurus nobilis* L. (Lauraceae); at the base of *Ceratonia siliqua* L. (Fabaceae).
*Botryobasidium conspersum* J. Erikss.	Sap	1			SPB: on dead branch of *Vitis vinifera* L. (Vitaceae).
*Brevicellicium olivascens* (Bres.) K.H. Larss. et Hjortstam	Sap			1	EKB: at the base of *Pteris* sp. (Pteridiaceae).
*Byssomerulius corium* (Pers.) Parmasto	Sap	1	1		SPB: at the base of *Hedera helix* L. (Araliaceae)/MSC: at the base of *Quercus acuta* Thunb. (Fagaceae).
*Ceratobasidium cornigerum* (Bourdot) D.P. Rogers	Sap		1	2	MSC: at the base of *Cordyline banksii* Hook f. (Asparagaceae)/EKB: base of *Musa basjoo* Siebold et Zucc. ex Iinuma (Musaceae); base of *Prunus laurocerasus* L. (Rosaceae).
*Cerioporus varius* (Pers.) Zmitr. et Kovalenko	Sap	1			SPB: on dead part of deciduous tree.
*Ceriporia viridans* (Berk. et Broome) Donk	Sap			1	EKB: on dead branch of *Citrus × sinensis* (L.) Osbeck (Rutaceae).
*Clavulina cinerea* (Bull.) J. Schröt.	Myc	1			SPB: at the base and on the roots of alive *Nerium oleander* L. (Apocynaceae).
*Clavulina ornatipes* (Peck) Corner	Myc			1	EKB: on the roots of *Magnolia obovata* Thynb. (Magnoliaceae).
*Clavulinopsis aurantiocinnabarina* (Schwein.) Corner	Sap			1	EKB: on the roots of *Magnolia obovata* Thunb. (Magnoliaceae) and *Cryptomeria japonica* (Thunb. ex L.f.) D.Don (Cupressaceae).
*Coniophora arida* (Fr.) P. Karst.	Sap		1	2	MSC: at the base of *Ginkgo biloba* L. (Ginkgoaceae)/EKB: dry base of *Cereus repandus* Mill. (Cactaceae); at the base of *Ligustrum ovalifolium* Hassk. (Oleaceae).
*Coniophora puteana* (Schumach.) P. Karst.	Sap		2	3	MSC: at the base of *Pittosporum tobira* (Thunb.) W.T.Aiton (Pittosporaceae); dead part of *Cordyline pumilio* Hook.f. (Asparagaceae)/EKB: at the base of *Cupressus sempervirens* L. (Cupressaceae); dead base of Pteris sp. (Pteridiaceae); base of *Dendrobium nobile* Lindley (Orchidaceae).
*Cylindrobasidium evolvens* (Fr.) Jülich	Sap		2	5	MSC: broken branch of *Acca sellowiana* (O.Berg) Burret (Myrtaceae); at the base of fern (Pteridiaceae)/EKB: on dead stem of *Hedera helix* L. (Araliaceae); dead leave of *Agave americana* L. (Asparagales); dead branch of *Bougainvillea spectabilis* Willd. (Nyctaginaceae); dead branch of *Hydrangea macrophylla* (Thunb.) Ser. (Hydrangeaceae); dead petioles of *Cycas revoluta* Thunb. (Cycadaceae).
*Flaviporus brownii* (Humb.) Donk	Sap	1			SPB: on dead deciduous tree and box wall (Bondartsev, 1953).
*Ganoderma applanatum* (Pers.) Pat.	Sap			1	EKB: at the base of *Coffea arabica* L. (Rubiaceae).
*Helicobasidium purpureum* (Tul.) Pat.	Pat		1		MSC: at the base of *Asparagus falcatus* (L.) Druce (Asparagales).
*Hymenochaete rheicolor* (Mont.) Lév.	Sap		1		MSC: at the base of *Sparmannia africana* L.f. (Malvaceae).
*Hymenochaete tabacina* (Fr.) Lev.	Sap	1	1		SPB: at the base of deciduous tree/MSC: on dead part of *Citrus reticulata* Blanco (Rutaceae).
*Hyphodontia arguta* (Fr.) J. Erikss.	Sap	1	2	3	SPB: on dead branch of *Citrus × sinensis* (L.) Osbeck (Rutaceae)/MSC: on the leaves of *Sabal palmetto* (Walter) Lodd. ex Schult. et Schult.f. (Arecaceae) ; at the base of *Ficus lyrata* Warb. (Moraceae)/EKB: dead part of *Monstera deliciosa* Liebm. (Araceae); on rotten part of *Lepidozamia peroffskyana* Regel (Zamiaceae); on dead stem of *Hedera helix* L. (Araliaceae).
*Hypochniciellum ovoideum* (Jülich) Hjortstam	Sap			1	EKB: at the base of *Tetrastigma voinierianum* (Sallier) Pierre ex Gagnep. (Vitaceae).
*Lentinus brumalis* (Pers.) Zmitr.	Sap	1	1		SPB: dead branch of *Laurus nobilis* L. (Lauraceae)/MSC: at the base of *Acca sellowiana* (O.Berg) Burret (Myrtaceae).
*Leptosporomyces raunkiaeri* (M.P. Christ.) Jülich	Sap			1	EKB: at the base and dead roots of *Chamaerops humilis* L. (Arecaceae).
*Lindtneria trachyspora* (Bourdot et Galzin) Pilát	Sap		1		MSC: at the base of *Drimys granadensis* L.f. (Winteraceae).
*Lyomyces crustosus* (Pers.) P. Karst.	Sap			4	EKB: on dead stem of *Hedera helix* L. (Araliaceae); at the base of Pteris sp. (Pteridiaceae); dead stem of *Monstera deliciosa* Liebm. (Araceae); dead base of *Aspedistra lurida* Ker Gawl. (Asparagaceae).
*Lyomyces erastii* (Saaren. et Kotir.) Hjortstam et Ryvarden	Sap		2	3	MSC: on dead leaf of *Washingtonia robusta* H. Wendl. (Arecaceae); on dead stem of *Vitis vinifera* L. (Vitaceae)/EKB: dead branch of at the base of *Ginkgo biloba* L. (Ginkgoaceae); dead root of *Philodendron bipinnatifidum* Schott ex Endl. (Araceae); on dead branch of *Psidium guajava* L. (Myrtaceae).
*Lyomyces sambuci* (Pers.) P. Karst.	Sap	1	3	6	SPB: at the base of deciduous tree/MSC: on dead branch of *Elaeodendron capense* Eckl. et Zeyh. (Celastraceae); dead branch of *Citrus maxima* (Burm.) Merr. (Rutaceae); dead base of *Clivia miniata* (Lindl.) Regel. (Amaryllidaceae)/EKB: On dead twig of *Citrus medica* L. (Rutaceae); on dead branch of *Acca sellowiana* (O.Berg) Burret (Myrtaceae); at the base of *Ficus carica* L. (Moraceae); at the base of *Musa maclayi* F.Muell. ex Mikl.-Maclay (Musaceae); on dead petioles of *Blechnum brasiliense* Desv. (Blechnaceae); on dead base of *Strelitzia reginae* Banks ex Aiton (Strelitziaceae).
*Microporus xanthopus* (Fr.) Kuntze	Sap			1	EKB: on woody box made of East-Asian *Carpinus cordata* Blume (Betulaceae) where grew *Magnolia obovata* Thunb. (Magnoliaceae).
*Peniophora cinerea* (Pers.) Cooke	Sap	1	3	5	SPB: on twig of deciduous tree/MSC: on branch of *Laurus novocanariensis* Rivas Mart., Lousa, Fern. Prieto, E.Diaz, J.C.Costa et C.Aguiar (Lauraceae); on dead branch of *Eucalyptus oblique* L’Her. (Myrtales); on dead part of *Paphiopedilum haynaldianum* (Rchb.f.) Stein (Orchidaceae)/ EKB: on dead stem of *Vitis vinifera* L. (Vitaceae); on dead twig of *Citrus limon* (L.) Osbeck. (Rutaceae); on dead branch of *Ficus carica* L. (Moraceae); on dead stem of *Aloe arborescens* Mill. (Asphodelaceae); on dead stem of *Heptapleurum arboricola* Hayata (Araliaceae).
*Physalacria cryptomeriae* Berthier et Rogerson	Sap			1	EKB: on dead twigs of *Cryptomeria japonica* (Thunb.ex L.f.) D.Don (Cupressaceae).
*Physalacria orientalis* (Kobayasi) Berthier	Sap			2	EKB: on dead branches and rotten base of *Magnolia grandiflora* L. (Magnoliaceae); at the base of *Grevillea robusta* A.Cunn. ex R.Br. (Proteaceae).
*Piloderma byssinum* (P. Karst.) Jülich	Myc		2	3	MSC: at the base of *Ocotea foetens* (Aiton) Baill. (Lauraceae); dead branch of *Nerium oleander* L. (Apocynaceae)/EKB: at the base of *Cryptomeria japonica* (Thunb.ex L.f.) D.Don (Cupressaceae); at the base of *Phoenix roebelenii* T.Anderson (Arecaceae); on the rotten base of *Iris japonica* Thunb. (Iridaceae).
*Ramariopsis minutula* (Bourdot et Galzin) R.H. Petersen	Sap		1		MSC: on the roots of *Dicksonia fibrosa* Colenso (Dicksoniaceae).
*Ramariopsis kunzei* (Fr.) Corner	Sap	1	1	2	SPB: on the roots of *Blechnum brasiliense* Desv. (Blechnaceae)/MSC: on the roots and at the base of *Olearia tomentosa* (J.C.Wendl.) DC. (Ateraceae)/EKB: on the roots and at the base of *Abutilon darwinii* Hook.f. (Malvaceae); on the roots of *Ilex crenata* Thunb. (Aquifoliaceae).
*Rhizoctonia fusispora* (J. Schröt.) Oberw., R. Bauer, Garnica et R. Kirschner	Sap		1	2	MSC: on the roots and bark of *Petrea volubilis* L. (Verbenaceae)/EKB: at the base of *Taxus baccata* L. (Taxaceae); on the soil and roots of *Casuarina equisetifolia* L. (Casuarinaceae).
*Rigidoporus lineatus* (Pers.) Ryvarden	Sap			1	EKB: at the base of *Eucalyptus* sp. (Myrtaceae) and the wall of a clay pot brought in 1958 from Batumi Botanical Garden. Fruitbodies were collected in 2002.
*Scopuloides rimosa* (Cooke) Jülich	Sap	1			SPB: at the base of deciduous tree.
*Sistotrema brinkmannii* (Bres.) J. Erikss.	Sap			2	EKB: at the base of *Casuarina equisetifolia* L. (Casuarinaceae); at the base of *Musa basjoo* Siebold and Zucc. ex Iinuma (Musaceae).
*Sertulicium niveocremeum* (Höhn. et Litsch.) Spirin et K.H. Larss.	Sap			2	EKB: at the base of *Grevillea robusta* A.Cunn. ex R.Br. (Proteaceae); at the base of *Cinnamomum verum* J.Presl. (Lauraceae).
*Subulicystidium longisporum* (Pat.) Parmasto	Sap	1			SPB: on rotten part of *Vitis vinifera* L. (Vitaceae).
*Thanatephorus terrigenus* (Bres.) G.Langer	Sap		1		MSC: at the base of *Meryta sinclairii* (Hook.f.) Seem. (Araliaceae).
*Thelephora palmata* (Scop.) Fr.	Myc/Pat		1		MSC: at the base and roots of *Myrsine africana* L. (Primulaceae).
*Thelephora terrestris* Ehrh. ex Fr.	Myc/Pat	1			SPB: at the base of deciduous tree.
*Tomentella cinerascens* (P. Karst.) Höhn. et Litsch.	Myc		1		MSC: at the base of *Acca sellowiana* (O.Berg) Burret (Myrtaceae).
*Tomentella lapida* (Pers.) Stalpers	Myc			1	EKB: at the base of *Magnolia grandiflora* L. (Magnoliaceae).
*Tomentella radiosa* (P. Karst.) Rick	Myc			2	EKB: at the base of *Manihot esculenta* Crantz. (Euphorbiaceae); at the base of *Musa basjoo* Siebold et Zucc. ex Iinuma (Musaceae).
*Trametes ochracea* (Pers.) Gilb. et Ryvarden	Sap		1	2	MSC: at the base of Uncarina grandidieri (Baill.) Stapf (*Pedaliaceae*)/EKB: at the base of *Ginkgo biloba* L. (Ginkgoaceae); at the base of *Ficus carica* L. (Moraceae).
*Trechispora nivea* (Pers.) K. H. Larss.	Sap	1			SPB: at the base of fern.
*Trechispora farinacea* (Pers.) Liberta	Sap	1	1	2	SPB: at the base of *Dicksonia fibrosa* Colenso (Dicksoniaceae)/MSC: at the base of *Eriobotrya japonica* (Thunb.) Lindl. (Rosaceae)/EKB: at the base of *Myriocarpa cordifolia* Liebm. (Urticaceae); on the dead base of *Zingiber officinale* Roscoe (Zingiberaceae).
*Typhula micans* (Pers.) Berthier	Sap		1	2	MSC: at the dead base of *Dianella* sp. (Asphodelaceae)/EKB: on dead part of *Brasiliopuntia brasiliensis* (Wild.) A.Berger (Cactaceae); on dead stem of *Sonchus canariensis* (Sch. Bip.) Boulo (Asteraceae).
*Xylodon asper* (Fr.) Hjortstam et Ryvarden	Sap			2	EKB: at the base of *Agave tequilana* F.A.Webber (Asparagaceae); at the base of *Tetrastigma voinierianum* (Sallier) Pierre ex Gagnep. (Vitaceae).
*Xylodon detriticus* (Bourdot) K.H. Larss., Viner et Spirin	Sap			3	EKB: on dead base of *Strelitzia nicolai* Regel et K.Koch (Strelitziaceae); at the base of *Ficus carica* L. (Moraceae); on dead base of *Ruscus aculeatus* L. (Asparagaceae).
*Xylodon flaviporus* (Berk. et M.A. Curtis ex Cooke) Riebesehl et Langer	Sap		2	5	MSC: on dead parts of *Distilium racemosum* Siebold et Zucc. (Hamamelidaceae); at the base of *Barberis rotundifolia* (Rosaceae)/EKB: on dead branch of *Carica papaya* L. (Caricaceae); on dead part of *Cereus repandus* Mill. (Cactaceae); on dead branch of *Nolina recurvata* Lem. (Asparagaceae); on dead stem of *Ficus elastic* Roxb. ex Hornem. (Moraceae); at the base of *Cycas revoluta* Thunb. (Cycadaceae).
*Xylodon raduloides* Riebesehl et Langer	Sap			2	EKB: on dead roots of *Monstera deliciosa* Liebm. (Araceae); on dead stem of *Pancratium* L. (Amaryllidaceae).
Total:		17	39	84	

Note: SPB—Saint Petersburg; MSC—Moscow; EKB—Ekaterinburg; Sap—saprobes; Myc—mycorrhizal; Pat—pathogens.

**Table 3 jof-09-01116-t003:** Aphyllophoroid fungi number of specimens collected on woody and herbaceous plants in the three urban greenhouses (number of specimens/%).

Parameter		Fungal Trophic Group			Total
		Saprobes	Pathogens	Mycorrhizal	
Plants	Woody	102/72.8	4/2.8	9/6.4	115/82.1
	Gymnosperms	9/6.4	0/0	0/0	9/6.4
	Angiosperms	**66/14.1**	2/1.4	**8/5.7**	75/53.6
	Arborescent	**27/19.3**	2/1.4	1/0.7	30/21.4
	Herbaceous	25/17.8	0/0	0/0	25/17.8
	Evergreen	15/10.7	0/0	0/0	15/10.7
	Ferns	10/7.1	0/0	0/0	10/7.1
	Annual	0/0	0/0	0/0	0/0
Total		127/90.7	4/2.8	9/6.4	140/100

Note: If a fungus belongs to several trophic groups, then the species is taken into account in all corresponding trophic groups in the table. The largest parameters are highlighted in **bold**.

**Table 4 jof-09-01116-t004:** Parameters of five plots along anthropogenic gradient in Ekaterinburg and surroundings.

Parameter	Plot				
	Greenhouses	Arboretum of Botanical Garden UrB RAS	City Center	Suburbs Forests	Old-Growth Forests
Number of sites	8	20	20	20	20
Average size of site, ha	0.08	0.08	0.08	0.08	0.08
Total size, ha	0.61	1.55	1.62	1.51	1.65
Share of non-native tree species, %	100	88 (100–72)	73 (100–48)	42 (79–26)	0
Total number of fungal species	39	86	43	64	68
Average number of fungal species per site	5.9 (2–8)	18.3 (13–25)	7.7 (4–12)	15.8 (12–21)	18.1 (11–24)
Menhinick Index	4.19	3.53	4.02	4.99	5.58
Shannon Index	3.48	3.31	3.54	3.88	4.42
Jaccuard Index	0.08 (0–0.24)	0.38 (0.25–0.46)	0.32 (0.24–0.40)	0.57 (0.49–0.68)	0.62 (0.58–0.83)
Whittaker Index	5.6	3.7	4.6	3.0	2.7
Morisita-Horn Index	0.04	0.19	0.18	0.24	0.30
Total number of non-native fungal species	8	9	3	1	0
Share of non-native fungal species, %	20.5	9.5	6.9	1.6	0
Species richness of non-native fungi, ha	13.1	5.8	1.8	0.7	0

Shown in parentheses is min.–max.

## Data Availability

Additional data will be provided upon request.

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
