# Peer review of "How Poor Is Aphyllophoroid Fungi Diversity in the Boreal Urban Greenhouses of Eastern Europe?"

_jof, 2023, doi:10.3390/jof9111116_

Round 1

Reviewer 1 Report

Comments and Suggestions for Authors

It is an interesting article on a topic still in development so it becomes a good contribution.

All my comments and suggestions are indicated directly in the text

Comments on the Quality of English Language

as above observed 

Author Response

Answers to questions from reviewer No. 1.

The authors of the manuscript are deeply grateful to the respected reviewer for his careful reading of the manuscript. We agree with all proposals.

There are only two points that can be discussed:

1) The reviewer suggests replacing the words poor with rich in the title of the manuscript. If we accept the position of the respected reviewer, then this distorts the meaning of the title (and the entire work). In our opinion, and the result presented in the manuscript, it is possible to even further strengthen the position of the word poor or scare - Is aphyllophoroid fungi diversity in the boreal urban greenhouses of Eastern Europe really so scarce? This is due to the fact that it is traditionally believed that the mycobiota in greenhouses is poor. Therefore, we tried to answer the question - how low is diversity... It turned out, yes, the number of species is low... however, diversity is not only the total number of species.

Therefore, if the respected reviewer agrees with our above-described arguments, we would like to keep the original title.

2) Also, replace the term saprob with saprotrophic. These words are synonyms, but over the past 10 years, the term saprob has been used in major scientific mycological monographs and articles. If we submit our manuscripts for review to mycological or environmental journals from Springer or Elsevier, the reviewers always indicate that it is preferable to write the “currently generally accepted” term saprob. In this regard, we also use this term.

Therefore, if the respected reviewer agrees with our above-described arguments, we would like to retain the original term saprob.

The reviewer also asks the question: In recent years, with the introduction of molecular analyses, it has resulted in several "well-known" taxonomic groups constituting different taxes. Please enter a few lines where you comment on how you treated your species without a molecular study.

Our answer: In our studies, we adhered to a classical morphological approach, based on the idea that the development of fruitbodies marks the accumulation of mycelium biomass within the substrate, i.e. predominance of this species in the corresponding substrate entity. We agree that molecular screening can reveal greater species diversity per substrate entity, but it often also reveals inactive propagules and ephemeral mycelia of low competitive potential, i.e. incapable of its complete colonization. By: Novozhilov Yu. K., Malysheva V. F., Malysheva E. F., Shchepin O. N., Azarov D. V., Zmitrovich I. V., Volobuev S. V., Kovalenko A. E. Hidden diversity of fungi and fungus-like protists in nature ecosystems: problems and prospects // Biosphere. 2016. V. 8, no. 2. pp. 202–215.

The reviewer asks the question: Thelephora terrestris a micorrhizal species?

Our answer: Yes, we are agree with your information, but it is also a pathogen. That's why in the previous sentence we wrote that it is a pathogen (facultative). In Russia, this species is one of the most aggressive pathogens in garden centers where 1-3 year old seedlings of pine, maple, oak, for example, are grown.

The reviewer also asks the question: Maybe I got lost, but I don't think I've seen what the average percentage of a natural area is. In this way I would have felt in a comparison with a result of 3 times lower.

Our answer: according to the data in Table 4, the average number of fungal species within a greenhouse’s site is 3.0 times lower compared to natural conditions... This means that in greenhouses on a site of 0.08 ha, on average, 3 times fewer species of aphyllophoroid fungi were identified, according to compared to natural old-growth forests on an site of 0.08 ha (5.9 species vs. 18.1 species).

As suggested by the reviewer, we have replaced the term alien with non-native. Corrected grammatical errors.

Reviewer 2 Report

Comments and Suggestions for Authors

This paper deals with fungal diversity in an urban greenhouse environment in the boreal region. Information on fungal diversity in various environments is of great significance in collecting and investigating basic data.

Several suggestions are made for the publication of this paper.

1. Compared to nature, the greenhouse environment has limited conditions, so it is natural that fungal diversity is low. However, the introduction needs to explain why verification of what is already known is necessary, the need for research, and the areas of application.

2. Since there is no method for the identification of fungi, separate descriptions of materials and methods are required.

3. If only fruiting bodies were identified for fungal diversity, a reasonable explanation is needed in the title or content as to why only fruiting body diversity was analyzed.

4. Tables showing fungal diversity require an indication of total population numbers. We need to make sure that there is enough volume to enable readers to discuss diversity.

If you rearrange the following, more meaningful results will be derived from the conclusion.

Author Response

Answers to questions from reviewer No. 2.

The authors of the manuscript are deeply grateful to the respected reviewer for careful reading of the manuscript. We agree with all proposals.

Below are our responses to all the reviewer’s suggestions:

  1. Compared to nature, the greenhouse environment has limited conditions, so it is natural that the diversity of mushrooms is small. However, the introduction should explain why it is necessary to test what is already known, the need for research and areas of application.

Our response: We added a sentence in the INTRODUCTION section to why we should test this hypothesis.

  1. Since there is no method for identifying fungi, separate descriptions of materials and methods are necessary.

Our answer: we added citations of the keys we used, what microscopes we used, and what museums the collected samples are stored in.

  1. If only fruiting bodies were identified for fungal diversity, a reasonable explanation must be provided in the title or content as to why only fruiting body diversity was analyzed.

Our answer: we have added information that explains why we only studied the fruiting bodies (basidiomata) of fungi in greenhouses.

  1. Tables showing fungal diversity require an indication of the total population size. We need to provide enough space to allow readers to discuss diversity.

Our answer: in Table 2 we added a line indicating the total number of mushroom samples collected in the greenhouses of each of the three cities.

Our answers are highlighted in pink.

Round 2

Reviewer 2 Report

Comments and Suggestions for Authors

The author faithfully responded to review comments on the manuscript and revised it, so it is judged to be sufficient for publication.